# Transcriptome analysis of the growth performance of hybrid mandarin fish after food conversion

**Wen-Zhi Guan**[1,2], **Gao-Feng Qiu**[1☯]*, **Feng-Liu**[2☯]*

**1** Key Laboratory of Freshwater Aquatic Genetic Resources, Ministry of Agriculture, National Demonstration Center for Experimental Fisheries Science Education, Shanghai Engineering Research Center of Aquaculture, Shanghai Ocean University, Shanghai, China, **2** Shanghai Fisheries Research Institute, Shanghai Fisheries Technical Extension Station, Shanghai, China

☯ These authors contributed equally to this work.
* gfqiu@shou.edu.cn (GFQ); lf6475@sina.com (FL)

**Data Availability Statement:** The RNA sequencing data for this study can be found in the Short Read Archive (SRA) of the National Center for Biotechnology Information (NCBI) under the

## Abstract

During recent years, China has become a hotspot for the domestication of mandarin fish, and this is of great commercial value. Although the food preference of domesticated mandarin fish has been studied, little is known about genes regulating their growth. We raised hybrid mandarin fish on artificial feed for 3 months, the results showed that the survival rate of hybrid mandarin fish was 60.00%. Their total length and body weight were 18.34 ±0.43 cm and 100.44 ±4.87 g. The absolute length and weight gain rates were 0.14 cm/d and 1.08 g/d, respectively. Finally, RNA sequencing (RNA-Seq) was performed to identify potential genes and pathways activated in response to growth performance. The transcriptome analysis generated 68, 197 transcripts and 45,871 unigenes. Among them, 1025 genes were up-regulated and 593 genes were down-regulated between the fast- and slow-growth fish. Finally, we obtained 32 differentially expressed genes, which were mainly related to fatty acid biosynthesis (e.g. *FASN* and *ACACB*), collecting duct acid secretion (e.g. *ATP6E* and *KCC4*), cell cycle (e.g. *CDC20* and *CCNB*), and the insulin-like growth factor (IGF) system (*IGFBP1*). These pathways might be related to the growth of hybrid mandarin fish. In addition, more potential single nucleotide polymorphisms (SNPs) were detected in the fast-growth fish than in the slow-growth fish. The results suggest that the interaction of metabolism and abundant alleles might determine the growth of hybrid mandarin fish after food conversion.

## Introduction

Mandarin fish, a delicacy, is an economically important fish. It is also one of the four popular freshwater fish and very popular in China. *Siniperca chuatsi* and *Siniperca scherzeri* are major mandarin fish. *S. chuatsi* grows fast, but it is easily prone to infection during culture. Contrarily, *S. scherzeri* has superior characteristics of strong disease resistance and easy domestication; they can be fed artificial feeds. However, they grow slowly. During recent years, China

accession numbers PRJNA610675. Online at https://www.ncbi.nlm.nih.gov/sra/PRJNA610675.

**Funding:** This work was supported by the Youth Talent Development Plan of Shanghai Municipal Agricultural System, China (Grant No. 20180303), and the Shanghai Agriculture Applied Technology Development Program, China (Grant No. Z20170204). This funding body has no role in the design of the study, collection, analysis, interpretation of data or in writing the manuscript.

**Competing interests:** The authors have declared that no competing interests exist.

has become a hotspot for the domestication of mandarin fish [1, 2]. In the wild, mandarin fish feed on certain live fish, and this leads to some serious problems in the artificial culture of mandarin fish. For example, feeding live fish to mandarin fish is very expensive and labour intensive. Additionally, these fish can be a source of pathogens or parasites, which can lead to a significant loss of cultured mandarin fish [3]. To overcome this problem, we used *S. chuatsi* and *S. scherzeri* to develop a hybrid F1 generation in 2010, and then backcrossed it with *S. scherzeri* three times and bred for two generations. Finally, the fifth generation fish was back-crossed with *S. chuatsi* to obtain a new hybrid mandarin fish [4]. During domestication, we found that the new hybrid mandarin fish has a faster growth rate and stronger disease resistance than the F1 hybrid and its parents, and it was easier to domesticate [4]. Furthermore, there were obvious growth differences in the hybrid fish after months of domestication. Therefore, genetic breeding of mandarin fish is essential to study the effect of domestication on growth and the underlying mechanism in the future.

During recent years, RNA sequencing (RNA-Seq) has been widely applied in fundamental research as an effective tool and the standard for transcriptome analysis. Transcriptome studies can holistically elucidate the functions and structures of genes and the molecular mechanisms of biological processes. The transcriptome is widely used to study the following: cold adaptation and thermal response of polar ectothermic species [5]; genetic diversity [6] and genetic management of commercially important fish [7]; genetic network regulating low-temperature tolerance in fish [8]; high-altitude adaptation [9] and the skeletal muscle of swimming fish [10]; adaptive responses to hypoxia [11], pathogenesis of diseases, and functions of the immune system in fish [12]; and sexual plasticity in sex-changing fish [13].

In the present study, in order to increase understanding the molecular mechanism of growth differences caused by food conversion, transcriptome sequencing was used to analysis the growth performance of hybrid mandarin fish after food conversion. We believe that these resources can be screened to reasonably explain the molecular mechanism of growth of fish, and would greatly aid domestication and breeding programs for mandarin fish.

## Materials and methods

### Ethics statement

The fish were handled in accordance with the Guide for the Care and Use of Laboratory Animals of the National Advisory Committee for Laboratory Animal Research. The protocol was approved by the Institutional Animal Care and Use Committee (IACUC) of Shanghai Fisheries Research Institute, Shanghai, China (Protocol Number: SHFRI-ACU-2018033). All surgery was performed under eugenol anesthesia, and all efforts were made to minimize suffering.

### Experiment fish

S. *chuatsi* and S. *scherzeri* were obtained from Shanghai Fisheries Research Institute, shanghai, China. In May 2010, S. *scherzeri* ♂ and S. *chuatsi* ♀ were used to produce hybrid F1. In 2013, the hybrid F1 ♂ and S. *scherzeri* ♀ were used to produce hybrid F2. In 2014, the hybrid F2 ♂ and S. *scherzeri* ♀ were used to produce hybrid F3. In 2015, the hybrid F3 ♂ and S. *scherzeri* ♀ were used to produce hybrid F4. In 2017, the hybrid F4 ♂ and S. *scherzeri* ♀ were used to produce hybrid F5 [4]. In May 2018, the hybrid F5 and S. *chuatsi* were used as male and female parents to produce hybrid F6 (H6) by artificial production and insemination. The selection criteria of the broodstock were the same in the experiment, and the parents were all mature adults with small growth differences. When the yolk of the H6 mandarin fry was almost absorbed, we offered small-sized *Megalobrama amblycephala* fry as food to them. When the

H6 hybrid mandarin fish were 3 cm long, we began to domesticate them on an artificial feed (Qingdao Qihao Biotechnology Co., Ltd). Training was usually done in the morning and evening. During training, live prey fish were used as feed in the beginning, and then changed to less dynamic fish and finally to the artificial feed that contained 40% fish meat. The successful food transition took approximately 15 days. After that, some of the untamed individuals will be eliminated. When their body length reached about 5.5–6.0 cm, three net cages of size 2 m × 2 m × 1.2 m, with 200 fish of the same size per cage, were used to continue growth experiments. The fish were counted every month, and the feeding period was 90 days.

## Growth performance and sample collection

In order to evaluate the growth performance, the body weight (accuracy = 0.01 g) of fish was measured using an electronic balance. Furthermore, the main traits (accuracy = 0.1 mm) including total length, body length, body height, and body width were measured using a compass and ruler at the end of the training period. Additionally, based on the body weight, the trained fish were divided into two groups, fast- (F) and slow-growth (S) groups, with five fish per group (S1 Table). The 10 fish were euthanised with Eugenol (JINYUAN, Jiangxi, China), and then the liver was collected in RNAstore Reagent (TIANGEN, Beijing, China), according to the manufacturer's protocol.

## Total RNA extraction, library construction and sequencing

According to the manufacturer's protocol, the total RNA from the liver samples was extracted using TRIzol reagent (Invitrogen, Carlsbad, CA, USA) and purified using the RNeasy Mini Kit (Qiagen, Valencia, CA, USA). RNA quality was checked on 1% agarose gels. RNA purity was analysed using Nanodrop 2000 (Thermo Scientific, MA, USA). The Truseq™ RNA Sample Prep Kit (Illumina, USA) was used to construct a cDNA library, according to the manufacturer's instructions. Magnetic beads with oligo(dT) (Invitrogen, USA) were used to purify the mRNA from the total RNA. Fragmentation buffer was added to break the mRNA into short fragments of approximately 300 bp. The first strand cDNA was synthesised with a short mRNA fragment and random hexamers, and the second strand cDNA was subsequently synthesised using DNA polymerase I and RNase H, to obtain a stable double-chain structure. The adaptor-ligated, size-selected cDNA were amplified by polymerase chain reaction (PCR) (15 cycles) and isolated on 2% agarose gels to construct the final cDNA library. Before sequencing, TBS-380 Picogreen (Invitrogen, USA) was used to quantify the DNA libraries. After cluster generation, the prepared libraries were sequenced on the Illumina Hiseq™ 4000 platform (Majorbio Biotech Co., Ltd., Shanghai, China), generating 2 × 150 bp paired-end reads.

## *De novo* assembly and functional annotation

In order to ensure the accuracy of the subsequent analysis results, we performed quality control of the original reads. Adapter sequences, >10% poly-N reads, and low quality reads were removed, to generate clean reads. All clean reads were then assembled using Trinity [14]. TransRate [15] and CD-HIT [16] were used to optimise and filter the assembled data. Subsequently, BUSCO [17] was used to evaluate the final sequences, which were considered unigenes or transcripts. Finally, the unigenes with E-value $< 10^{-5}$ were compared against those in the Non-Redundant Protein (NR), SwissProt, Pfam, Clusters of Orthologous Group (COG), Gene Onthology (GO), Kyoto Encyclopedia of Genes and Genomes (KEGG) databases to obtain annotation information, and then counted them.

## Differential expression and functional enrichment analyses

To analyse the differentially expressed genes (DEGs) among different samples, RSEM [18] was used to quantify the unigenes (Anders and Huber, 2010). DESeq2 was used to perform differential gene expression analysis. Genes identified using DEGSeq2 with an adjusted P-value < 0.05 were considered DEGs [19]. The GO and KEGG pathway enrichment analyses were performed on the DEGs, using Goatools software [20] and R program, respectively.

## Detection of simple sequence repeats and single nucleotide polymorphisms

To identify candidate single nucleotide polymorphisms (SNPs) from the assembled transcripts of all samples, SAMtools (http://samtools.sourceforge.net/) and BCF tools (https://github.com/samtools/bcftools) were used MISA (http://pgrc.ipk-gatersleben.de/misa/) was used for simple sequence repeat (SSR) detection in unigenes with the default parameters.

## Quantitative real-time PCR

Quantitative real-time (qRT)-PCR was used to verify the reliability of the RNA-Seq results. RNA used for transcriptome sequencing was converted into cDNA using the PrimeScript RT Reagent Kit (Takara, Japan). The cDNA was then used for the qRT-PCR analysis using specific primers (Table 1). β-actin was used as the control mainly because of its stability. qRT-PCR was performed with SYBR Green Premix ExTaq (TaKaRa, Japan) using the CFX96 Touch™ real-time PCR Detection System (BioRad, USA). All the samples were analysed in triplicate, and fold changes in gene expression were calculated using the $2^{-\Delta\Delta CT}$ method [21].

## Statistical analysis

Data pertaining to growth performance were analysed using SPSS 19.0 software, and one way analysis of variance (ANOVA) and significance test were performed. The results are expressed as mean ± standard deviation (SD). Significance was accepted at the level of $P < 0.05$.

# Results

## Growth performance

The growth performance of the hybrid mandarin fish is shown in Fig 1. A comparison of data obtained at 30 and 90 days revealed that with the increase in domestication period, the survival rate of hybrid fish gradually decreased (Fig 1A). At the end of the training period, the survival rates of the fish was 60.00% (S2 Table). Before domestication, the mean total length (TL) and mean body weight (BW) were 5.82 ± 0.54 cm and 2.84 ± 0.65 g, respectively. After 3 months of breeding, their total length and body weight were 18.34 ± 0.43 cm and 100.44 ± 4.87 g (Fig 1C

**Table 1. Primer sequences used for real-time RT-PCR.**

| Gene name | Forward primer sequence (5'-3') | Reverse primer sequence (5'-3') |
|:---:|:---:|:---:|
| FASN | CGGTGACGCCACTCAACAAG | GCACTGCAGGCCATAGCAAG |
| ACACB | CTGGCCAAGATGGTGGGCTA | CCTGCAGGCGAGGATTCAGT |
| ACSBG | ATGTGTCACGGACGACAACG | CTCCGACACCTTGGCTGCT |
| ATP6E | TCGAGAAAGGTCGTCTGGTG | TCCAAGTCTTTGACGAGCCT |
| CDC20 | TTGGAGGGAAGCCGCTGAAT | AGTCCTCCTCGCGCTCTAAC |
| CCNB | CAGAGGTGCAGGTTCTGCCT | ATGGCCCGCATGTTACCAGT |
| IGFBP1 | CCCACATGATGCTCCCTTCT | CGCAGGAGAAACTGAACGAC |
| β-actin | ATCGCCGCACTGGTTGTTGAC | CCTGTTGGCTTTGGGGTTC |

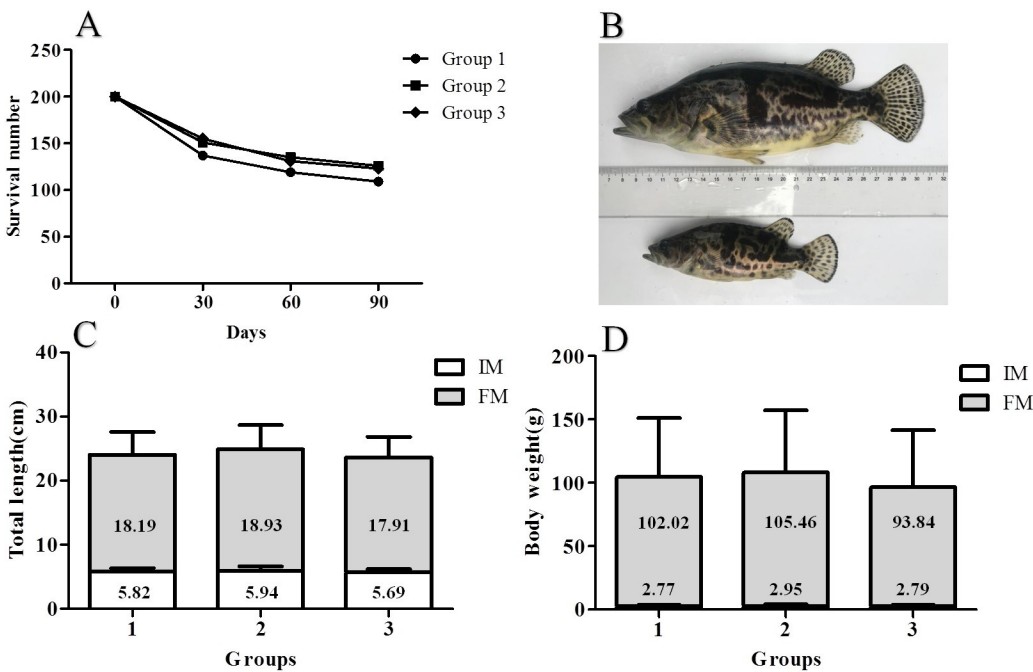

**Fig 1. The growth performance of hybrid mandarin fish.** (**A**) The survival number of hybrid mandarin fish. (**B**) The morphology of domesticated hybrid mandarin fish. (**C**) The change of total length of hybrid mandarin fish. (**D**) The change of body weight of hybrid mandarin fish. IM, Initial Measurement (0 day). FM, Final Measurement (90days).

and 1D). The absolute length and weight gain rates were 0.14 cm/d and 1.08 g/d. Furthermore, these fish vary greatly in weight when the fish were counted at the end of the experiment (Fig 1B and S1 Table).

## Sequencing and annotation of unigenes

After sequencing, 532.71 million paired-end reads (2 × 150 bp) and 80.44 billion bp were generated. After trimming and quality filtering, 526.65 million clean reads (98.86%) and 78.80 billion bp (97.96%) were obtained for the subsequent analyses. The number of clean reads per sample ranged from 46.12 to 56.58 million, with a mean of 52.67 million. The Q20 and Q30

**Table 2. Summary of sequences analysis.**

| Sample | Raw reads | Raw bases | Clean reads | Clean bases | Q20(%) | Q30(%) | GC content(%) |
|---|---|---|---|---|---|---|---|
| F1 | 46620884 | 7039753484 | 46118582 | 6906670060 | 97.98 | 93.95 | 49.18 |
| F2 | 57149480 | 8629571480 | 56580896 | 8470705187 | 97.99 | 93.99 | 49.60 |
| F3 | 55573022 | 8391526322 | 54997656 | 8228952151 | 97.99 | 94.00 | 49.36 |
| F4 | 55056242 | 8313492542 | 54491194 | 8154164513 | 98.08 | 94.22 | 49.47 |
| F5 | 56702058 | 8562010758 | 56105974 | 8393382060 | 98.04 | 94.13 | 49.55 |
| S1 | 49655296 | 7497949696 | 49037308 | 7337067610 | 97.76 | 93.46 | 49.69 |
| S2 | 53680550 | 8105763050 | 53094284 | 7942035364 | 97.93 | 93.85 | 49.57 |
| S3 | 55210500 | 8336785500 | 54653148 | 8176427560 | 97.81 | 93.56 | 49.74 |
| S4 | 51259504 | 7740185104 | 50524030 | 7559072542 | 97.78 | 93.51 | 50.11 |
| S5 | 51798836 | 7821624236 | 51051630 | 7635673150 | 97.83 | 93.63 | 50.18 |
| Total | 532706372 | 80438662172 | 526654702 | 78804150197 | --- | --- | --- |

values of all samples were approximately 98% and 94%, respectively, and the GC content was approximately 49%–50% (Table 2). An optimised assembly of the reads generated 68 197 transcripts and 45 871 unigenes, ranging from 201 to 24 264 bp, with an average length of 1472.99 bp. The N50 and E90N50 values of the obtained transcripts were 2324 and 2562 bp, respectively. Approximately 51% of the transcripts and 58% of the unigenes were distributed in the 0–1000 bp region (S1 Fig). A total of 45 871 unigenes were annotated against those in the six public databases (NR, Swiss-Prot, Pfam, COG, GO, and KEGG), using Blastx and Blastn. Among these unigenes, 21 039 (45.87%) were annotated in the NR, 17 911(39.05%) in the Swiss-Prot, 15 869 (34.59%) in the Pfam, 10 643(23.2%) in the COG, 6 187(13.49%) in the GO, and14 077(30.69%) in the KEGG databases.

In the GO analysis, 30 400 unigenes (66.27%) had GO annotation, with 24.43% for molecular functions, 37.01% for cellular components, and 38.55% for biological processes. They were further assigned to 46 functional terms, and binding (3249 unigenes), catalytic activity (2497 unigenes), and cellular process (2489 unigenes) represented the major terms (Fig 2). Functional classification of the KEGG pathways provided a valuable resource for investigating specific processes and pathways in the liver of the hybrid mandarin fish. We mapped the 22 828 unigenes to the reference canonical pathways in the KEGG database to identify the biological pathways related to growth. They were classified into six main categories, comprising 43 main pathways. Human diseases was the largest category (6844 unigenes, 29.98%), followed by Organismal Systems (5669 unigenes, 24.83%) (S2 Fig).

## Differential expression analysis and functional enrichment

We found a high number of unigenes with differential expression in the S vs. F groups. Differentially expressed genes with two-fold change in expression and adjusted $p$ value $< 0.05$ were identified. We identified 1618 DEGS in the liver of the S vs. F groups, with 1025 up- and 593

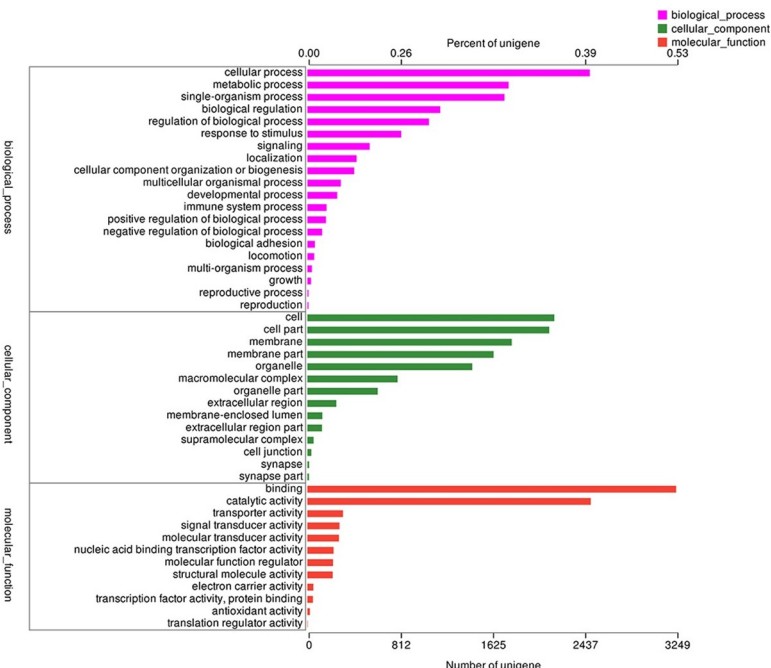

**Fig 2. Functional annotation of hybrid mandarin fish transcripts based on GO categorization.**

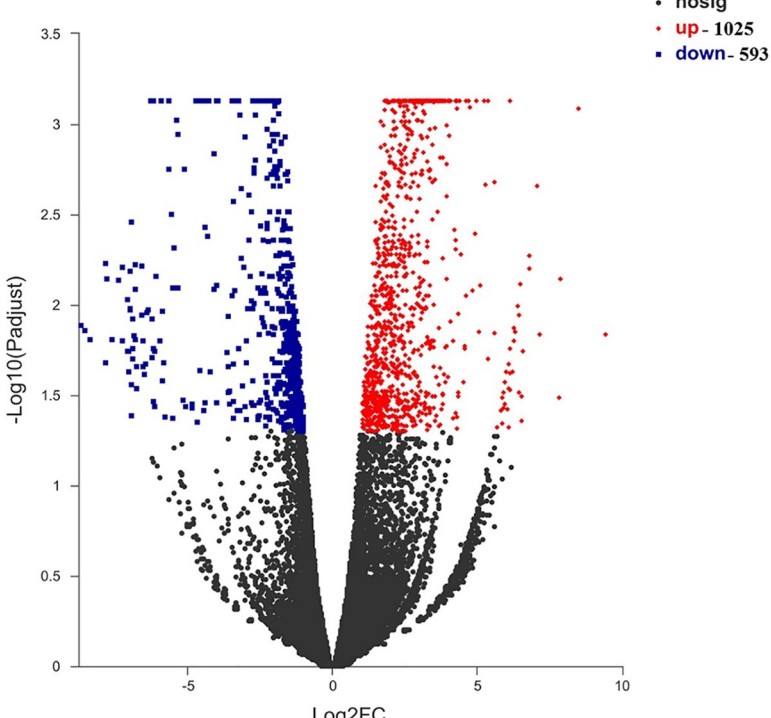

**Fig 3. DEGs analysis and volcano plot for fast- (F) and slow-growth (S) groups of hybrid mandarin fish.** The x-axis is the value of Log2 (Fold Change), and the y-axis is the value of -Log10 (padjust). The red and blue dots reveal the up- and down-regulated DEGs, the black dots reveal non-significantly genes.

down-regulated genes (Fig 3). The KEGG pathway enrichment analysis of the DEGs showed that there were 20 pathway in five categories, among which cell cycle was the most significant pathway in the S vs. F groups after domestication (S3 Fig). This was followed by cell cycle, collecting duct acid secretion and fatty acid biosynthesis. Additionally, we identified the IGF system gene (*IGFBP1*) in the S vs. F groups. In all, we obtained 32 differentially expressed genes in the S vs. F groups (Fig 4 and S3 Table).

## SNP and SSR discovery

A large number of valuable and polymorphic markers were obtained from the RNA-Seq analysis. We detected 2 311 424 potential SNPs in the F group and 2 214 210 potential SNPs in S group (S4 Table). Transitions were more frequent than transversions among the identified SNPs. The frequency of A/G (26.23%) and T/C (25.90%) was the highest in transition and the frequency of A/T (14.30%) and T/A (14.20%) was the highest in transversion (Fig 5A). Additionally, we found 22 399 potential SSRs in 14 873 unigenes (32.42%). Most of the SSRs were mono-nucleotide (44.78%) and di-nucleotide repeats (36.27%) (Fig 5B). The mainly type repeat number of SSRs was 6–10 nucleotides.

## qRT-PCR validation

The qRT-PCR was performed to validate the data obtained by RNA-Seq. Seven genes (namely, *ACACB*, *ACSBG*, *FASN*, *ATP6E*, *CCNB*, *CDC20*, and *IGFBP1*) from various categories were selected for confirmation. The results showed that the data obtained by qPCR were basically consistent with those of the RNA-Seq analysis (Fig 6).

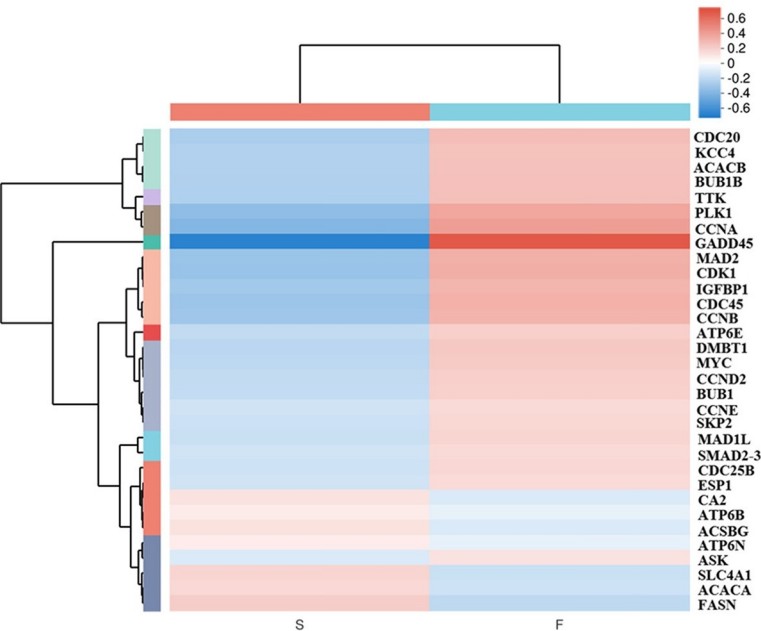

**Fig 4. Hierarchical cluster analysis of DEGs involved in the growth superiority.** Red and blue gradients indicate the up- and down- regulation of DEGs, respectively.

## Discussion

### Fish domestication

Research on domestication of animals has been carried out for several years [22]. Environmental conditions significantly alter hybrids from wild types, and this considerably affects the efficiency of domestication. Some studies on the mechanism of animal domestication have been conducted [23–25]. The domestication of fish with high commercial values can reduce labour force and cost, and also accelerate the development of aquaculture industry. So far, several fish have been domesticated, including some carnivorous fish [1, 26]; for example, *Elopichthys bambusa* [27], *Epinephelus malabaricus* [28], and *Lateolabrax japonicus* [29].

In recent years, some studies have been performed on the domestication for mandarin fish [1, 2, 30, 31]. It is confirmed that fish size, breeding density, starvation time, and artificial feed quality are closely related to the efficiency of mandarin fish culture [32]. In this study, the survival rates of mandarin fish was 60.00% at the end of the training period. The mortality of fish was mainly due to illness and swallow cause of individual differences. Here, we found that the growth performance of domesticated fish varied widely under the same feeding conditions, that would lead weak individuals are restricted by the strong individuals. Some behaviours in fish have a genetic basis, and therefore, the fish can respond to domestication selection. Different feeding conditions can lead to growth differences [1]. A previous study reported that sensory modality and associative learning appear to be critical factors in food conversion in fish [31]. Interestingly, domestication caused behaviour and brain size changes in clonal *Oncorhynchus mykiss* lines [25].

### Differentially expressed genes of growth

Growth trait is of great economic importance for all species used in aquaculture. Fast-growth fish are more profitable than slow-growth fish, as it can accelerate the production. Previous

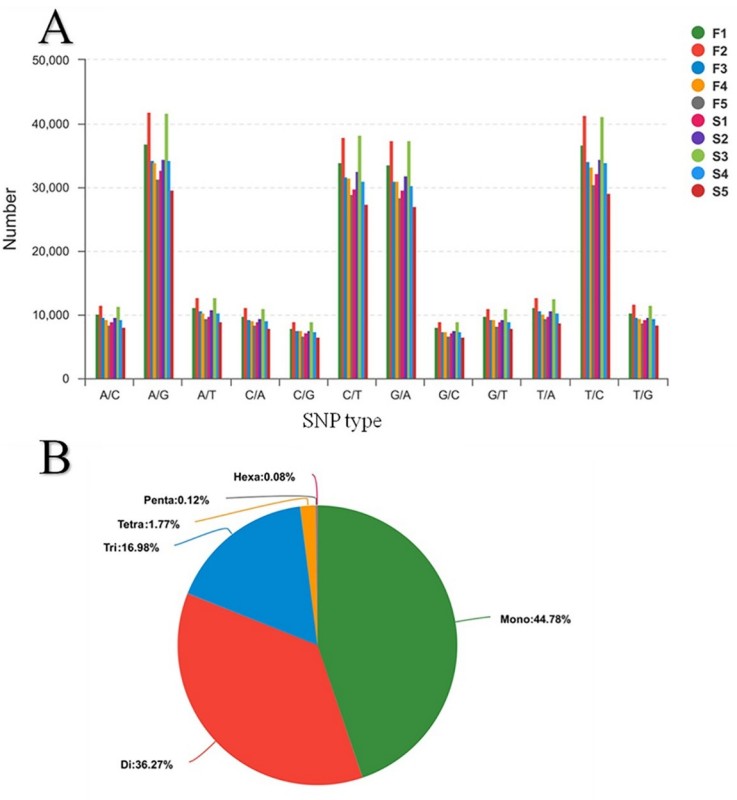

**Fig 5. A,** SNPs distribution in samples. The X-axis represents the type of SNP, and the y-axis represents the number of SNPs in different samples. **B,** the percentage of different SSR motifs. Different colors represent the number of different SSR types.

studies have revealed the unique mechanisms controlling food preference in the mandarin fish, involving retinal photosensitivity, circadian rhythm, appetite control, learning and memory, and SNP abundance [2]. Here, we found that the growth of hybrid mandarin fish varied greatly in the later stages during the feeding experiment for 3 months. RNA-Seq was

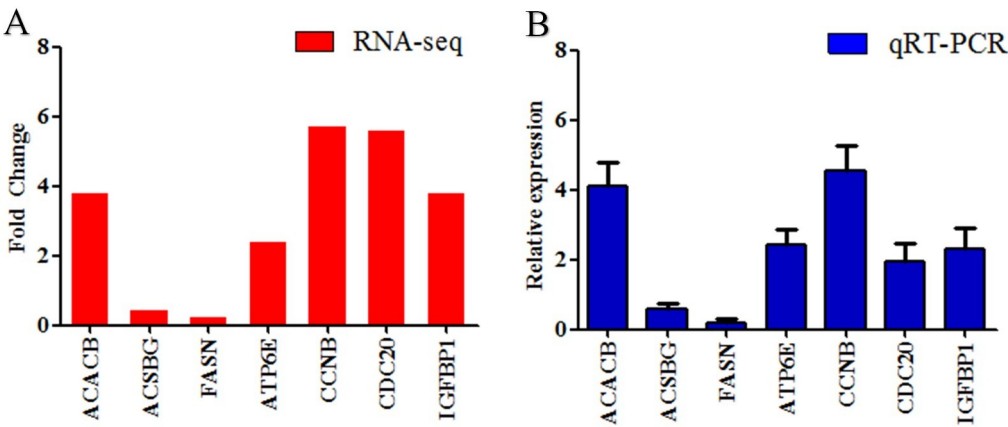

**Fig 6. Validation of RNA-seq data by qRT-PCR.** (A) The RNA-seq data. (B) The qRT-PCR data. The values are the ratio of gene expression levels in the fast-growth group to those in the slow-growth group by the two methods.

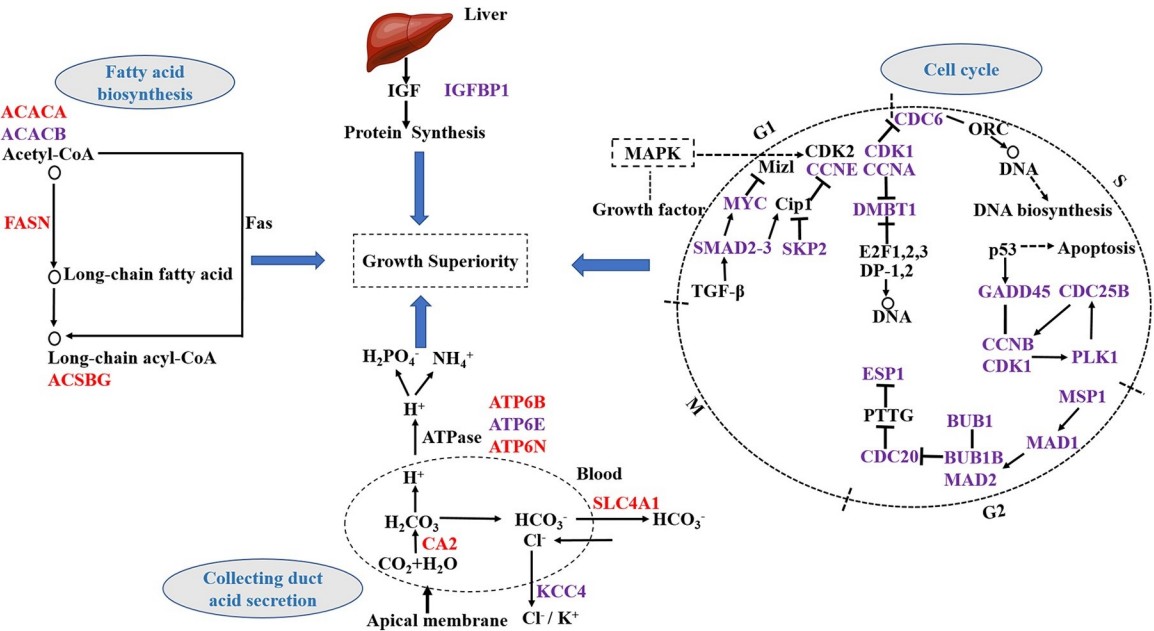

**Fig 7. The predicted map of DEGs regulated growth superiority.** Red and purple represent the up- and down-regulation of DEGs, respectively.

performed to identify genes associated with the difference in growth performance. We identified potential DEGs, mainly involved in fatty acid biosynthesis, collecting duct acid secretion, cell cycle and IGF system (Fig 7).

**Fatty acid biosynthesis.** In animals, fatty acids (FAs) play an important role in several physiological processes, and the majority of complex lipid molecules are derived by the transformation of FAs through several biochemical pathways. Acetyl-CoA carboxylase (ACACA) and fatty acid synthase (FASN) are crucial rate-limiting enzymes, and they play a crucial role in weight difference in the abdominal adipose tissue in growing animals [33]. Four genes involved in the FA biosynthesis pathway, namely, *ACACA*, *ACACB*, *ACSBG*, and *FASN*, were differentially expressed in the S vs. F groups of hybrid mandarin fish in our study. The highest expression of *ACACA* was observed in lipogenic tissues such as the liver and adipose tissue [34]. *ACACB* was significantly associated with milk yield, fat yield, fat percentage, protein yield, and protein percentage [35]. *ACSBG* is important for the lipid metabolism pathway in vertebrates [36]. *FASN* plays an important role in FA metabolism, and variation in *FASN* expression has been reported to be associated with both fat and meat traits [37]. Taken together, changes in the expression levels of these fatty acid biosynthesis-related genes in the F and S groups of mandarin fish might affect their growth.

**Collecting duct acid secretion.** H-K-ATPases are integral membrane proteins of the collecting duct, a segment critical for the final regulation of electrolyte excretion and bicarbonate (HCO₃) synthesis and reabsorption [38, 39]. It has been shown that H(+)-K(+)-ATPase mediates net acid secretion in rat terminal inner medullary collecting duct [40]. In animals, *CA2* can interconvert carbon dioxide and bicarbonate to maintain acid–base balance in the blood and other tissues, and to help transport carbon dioxide out of tissues (carbonic anhydrase-like isoform X2, CA2). *KCC4* mediates potassium and chloride exit from cells, and it might play an important role in salt absorption via the distal convoluted tubule [41]. *SLC4A1* plays a central role in $CO_2$ transport, and it is the major glucose transporter in human erythrocytes [42]. In

our study, the expression of *ATP6E* and *KCC4* was significantly higher in fast-growth fish than in slow-growth fish, potentially contributing to acid secretion by the collecting duct.

**Cell cycle.** A previous study has shown that *CCNB1/CDK1*-mediated phosphorylation provides cells the energy for G2/M transition and shortens the overall cell cycle duration, and thus, plays an important role in the cell cycle and proliferation [43]. *CDC20* can promote the proliferation of cutaneous squamous cell carcinoma [44]. Moreover, overexpression of *CDK1*, *CCNB1*, *CDC20*, *BUB1*, *MAD2L1*, and *BUB1B* in tumour tissues could increase cell proliferation and division [45]. *DMBT1* links mucosal defence and epithelial regeneration [46]. Esp1 expression is localised to the mitotic spindle during anaphase [47]. Innormal hepatocytes, *Gadd45β* facilitates cell survival, growth, and proliferation [48]. Polo-like kinase 1 (*Plk1*) is a serine/threonine kinase that is involved in the G2/M phase transition in the cell cycle in vertebrates [49]. *TTK* functions as a key regulator of spindle pole assembly, meiosis, DNA damage response, and cytokinesis [50]. In the present study, we found that the expression of 21 cell cycle-related genes was higher in the fast-growth fish than in the slow-growth fish; the up-regulation of these genes could influence cell proliferation and differentiation in mandarin fish.

**IGF system.** The IGF system plays a pivotal role in the regulation of growth. IGFBPs are important regulatory factors in the IGF system. They are mainly synthesised by the liver. Previous research has indicated that IGFBPs can inhibit and/or potentiate IGF activities depending on the physiological context [51, 52]. Under some conditions, *IGFBP1* can also enhance IGF function rather than inhibit it. For instance, *IGFBP1* potently promoted β-cell regeneration by potentiating α- to β-cell transdifferentiation in zebrafish [53]. Moreover, hypoxia significantly up-regulated hepatic *Igfbp1* expression in the liver [54]. In the present study, we observed a higher expression of *IGFBP1* in the fast-growth fish than in the slow-growth fish. Similar results have been reported in hybrid grouper, suggesting the gene related to the GH/IGF axis might play an important role in growth superiority of the hybrid fish [55].

## Single nucleotide polymorphisms and simple sequence repeats

RNA-Seq technology has been effectively applied to develop molecular markers in fish [6]. High-quality transcriptomes and a large set of gene-linked SNPs would facilitate functional and population genomic studies in endangered species [56]. Studies on molecular markers in mandarin fish have been conducted, including food habit domestication traits (SNP) [30], growth traits (SNP) [57] and SSR [58], and population genetic analysis (SSR) [59]. These studies were performed in *S. chuatsi*. Furthermore, 40 polymorphic SSR markers have been developed and characterised in *S. scherzeri* [60]. In the present study, numerous SNPs and SSRs were detected. The number of SNPs in the F group was significantly higher than that in the S group. It has been confirmed that abundant variant alleles are closely linked to food preference [2]. Moreover, the main types of SSRs were 6–10 mono and di-nucleotide repeats. These markers will be useful for diversity, mapping, and marker-assisted studies in mandarin fish (S1 File) [59].

## Conclusions

In the present study, new hybrid mandarin fish were raised for 3 months on an artificial feed containing 40% fish meat. The results showed that using appropriate methods can obtain better domestication effect, although the growth performance of domesticated fish varied widely under the same feeding conditions. The liver transcriptome analysis revealed that fatty acid biosynthesis, collecting duct acid secretion, cell cycle, IGF system, SNPs and SSRs were closely related to the growth of hybrid mandarin fish after food conversion. These results of transcriptome analysis will increase our understanding of the molecular mechanism involved in growth differences caused by food conversion. Future studies should focus on specific molecular

markers related to growth and domestication in order to accelerate the process of domestication and breeding of mandarin fish.

## Supporting information

**S1 Table. The detailed sample information of hybrid mandarin fish.**
(DOC)

**S2 Table. The domestication rate and survival rate of hybrid mandarin fish.**
(DOC)

**S3 Table. The data of DEGs in the S vs. F groups of hybrid mandarin fish.**
(DOC)

**S4 Table. The SNP types information of hybrid mandarin fish.**
(DOC)

**S1 File. The SNP information for growth-related genes of hybrid mandarin fish.**
(XLS)

**S1 Fig. Length distribution of transcripts / unigenes of hybrid mandarin fish.**
(TIF)

**S2 Fig. Functional annotation of hybrid mandarin fish transcripts based on KEGG categorization.**
(TIF)

**S3 Fig. The KEGG pathway enrichment analysis of the DEGs.** The y-axis represents the KEGG pathway, and the x-axis represents the significance level of enrichment. Different colors represent 7 branches of the KEGG metabolic pathway, containing cellular processes (CP), metabolism (M), organismal systems (OS), environmental information processing (EIP), human diseases (HD).
(TIF)

## Acknowledgments

We show appreciation to Shanghai Majorbio Company for the dynamic help provided during the analysis of the sequencing data.

## Author Contributions

**Conceptualization:** Gao-Feng Qiu,  Feng-Liu.

**Data curation:** Wen-Zhi Guan.

**Formal analysis:** Wen-Zhi Guan.

**Funding acquisition:** Wen-Zhi Guan,  Feng-Liu.

**Investigation:** Wen-Zhi Guan.

**Methodology:** Wen-Zhi Guan, Gao-Feng Qiu,  Feng-Liu.

**Project administration:**  Feng-Liu.

**Resources:** Wen-Zhi Guan,  Feng-Liu.

**Software:** Wen-Zhi Guan.

**Supervision:** Gao-Feng Qiu, Feng-Liu.

**Validation:** Wen-Zhi Guan.

**Visualization:** Wen-Zhi Guan, Gao-Feng Qiu, Feng-Liu.

**Writing – original draft:** Wen-Zhi Guan.

**Writing – review & editing:** Wen-Zhi Guan, Gao-Feng Qiu, Feng-Liu.

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
