## [Decision Letter · Decision Letter 0]

12 Aug 2020

PONE-D-20-17403

Transcriptome analysis of the growth performance of hybrid mandarin fish after food conversion

PLOS ONE

Dear Dr. Guan,

Thank you for submitting your manuscript to PLOS ONE. After careful consideration, we feel that it has merit but does not fully meet PLOS ONE’s publication criteria as it currently stands. Therefore, we invite you to submit a revised version of the manuscript that addresses the points raised during the review process.

1. The data cannot support the conclusions. PLOS ONE is designed to communicate primary scientific research,

and welcome submissions in any applied discipline that will contribute to the base of scientific knowledge. But

the data of this manuscript cannot support the conclusions.

2. This manuscript has the statistical analysis problem.

3. This manuscript needs to adhere the PLOS Data Policy. The authors need to make all methods, materials and

data underlying the findings in their manuscript fully available.

4. There are concerns needed to be addressed and some descriptions need to be clarified. The revised manuscript needs to address each of the comments of the reviewers.

We look forward to receiving your revised manuscript.

Kind regards,

Tzong-Yueh Chen, Ph.D.

Academic Editor

PLOS ONE

Journal Requirements:

Reviewers' comments:

Reviewer's Responses to Questions

**Comments to the Author**

1. Is the manuscript technically sound, and do the data support the conclusions?

Reviewer #1: Yes

Reviewer #2: Partly

2. Has the statistical analysis been performed appropriately and rigorously? 

Reviewer #1: N/A

Reviewer #2: I Don't Know

3. Have the authors made all data underlying the findings in their manuscript fully available?

Reviewer #1: Yes

Reviewer #2: No

4. Is the manuscript presented in an intelligible fashion and written in standard English?

Reviewer #1: Yes

Reviewer #2: Yes

5. Review Comments to the Author

Reviewer #1: The author wants to use the transcriptomic analysis to compare the differences in the molecular mechanism between individuals with different growth performance after domestication (food conversion), and to identify SNPs that may be related to this as a benchmark for subsequent selection. But the experimental design has many doubts as follows: First, the author should emphasize the selection criteria of the broodstock, otherwise the differences between the broodstock individuals may affect the representative meaning of the offspring. Second, in the process of domestication, the cannibalism might be occurred due to size selection of fish was not performed. Thus, the better growth performance might be due to eat more. Just like the author said the behavior might have a genetic basis, but the results are focused on growth performance instead of behavior. If the author wants to emphasize the growth performance, it is recommended that the experimental design should be rearrangement. For example, size sorting should be carried out during the domestication process, or the fish fry that has been domesticated with the same size should be selected for another growth experiment. In my opinion, the result will be more representative. There are, however a few inconsistencies that require further explanations or changes prior to publication in PLOS ONE.

Results:

Figure 1A and C: The result of 1A is not represented survival rate. The initial length of fish in Fig. 1C is between 5.69-5.92 cm, which is inconsistent of that mentioned in materials and methods.

Figure 6: The calculation method of fold change in RNA-seq and qRT-PCR is different. It is suggested that the FC results of two analyses should be separated. By the way, the FC results of CDC20 are different in these two analyses. In addition, “RNA-sep” should be changed to “RNA-seq”.

Furthermore, there are some typos in the indicated lines as follows:

Line 107, “De novo” should be italicized.

Line 132, There is an extra space between control and mainly.

Line 141, “Date” should be changed to “Data”.

Reviewer #2: The manuscript (PONE-D-20-17403) entitled "Transcriptome analysis of the growth performance of hybrid mandarin fish after food conversion" which submitted to PLOS ONE by Guan WZ et al. has been reviewed.

The author used artificial feed to raise hybrid mandarin fish for 3 months, and measured the food conversion rate, survival rate, total length, and body weight of the hybrid fish, absolute length and weight gain rate. Then use RNA-Seq to identify potential genes involved in the pathways activated growth performance. For example: fatty acid biosynthesis (FASN and ACACB), collection of duct acid secretion (ATP6E and KCC4), cell cycle (CDC20 and CCNB) and insulin-like growth factor (insulin) -like growth factor, IGF) system (IGFBP1) and other pathways may be related to the growth of mandarin fish. Finally, it was also found in fast-growth fish, more potential single nucleotide polymorphisms (SNPs) were detected than in slow-growing fish. However, there are still some problems with the experimental design and content of this manuscript, which need to be supplemented and further explained and clarified by the author.

Q1-

What is the breeding pattern and origin of “the fifth generation hybrid fish” mentioned in this manuscript material and method? Regarding the source and definition of breeding biological materials, it is necessary to ask the author to add more details in the manuscript.

Q2-

What is the control group for the measurement of food conversion rate, survival rate, total length, body weight, absolute length, and weight gain rate of hybrid fish? In theory, there should be non-hybrid pure breeds or hybrids of different generations for comparison. To use this to emphasize the advantages of hybrids, the author should add Siniperca chuatsi and Siniperca scherzeri two uncrossed pure breeds. It is the control group (triple group) and compares various growth and survival values.

Q3-

P156, Group1-3 in Figure 1(A) refers to the number of deaths at 0, 30, and 90 days in the triple-repeat population of the hybrid population. Why is there no 60-day data? It is impossible to understand why the author presented the triple-duplicate data of a hybrid species alone. What is the significance?

Q4-

Figure 1(C) and Figure 1(D) are the growth data of how many months old? It is suggested that the author should make up and give details.

Q5-

In order to make it easier for readers to understand, basically the author should provide the growth data or growth curve of the F and S groups at each month of age?

Q6-

Although the author mentioned in the manuscript that more potential single nucleotide polymorphisms (SNPs) are detected in fast-growth fish than in slow-growing fish. However, the author did not actually provide such comparative figures or tables in the manuscript.

Q7-

There are more potential single nucleotide polymorphisms (SNPs) found in fast-growing fish. These nucleotide variation site markers are mainly located on those important growth genes. The authors also found in this result. No relevant research diagrams or tables are provided in this manuscript.

6. PLOS authors have the option to publish the peer review history of their article (what does this mean?). If published, this will include your full peer review and any attached files.

Reviewer #1: No

Reviewer #2: No

---

## [Author Response · Author response to Decision Letter 0]

30 Aug 2020

Reviewer #1: The author wants to use the transcriptomic analysis to compare the differences in the molecular mechanism between individuals with different growth performance after domestication (food conversion), and to identify SNPs that may be related to this as a benchmark for subsequent selection. But the experimental design has many doubts as follows: First, the author should emphasize the selection criteria of the broodstock, otherwise the differences between the broodstock individuals may affect the representative meaning of the offspring. Second, in the process of domestication, the cannibalism might be occurred due to size selection of fish was not performed. Thus, the better growth performance might be due to eat more. Just like the author said the behavior might have a genetic basis, but the results are focused on growth performance instead of behavior. If the author wants to emphasize the growth performance, it is recommended that the experimental design should be rearrangement. For example, size sorting should be carried out during the domestication process, or the fish fry that has been domesticated with the same size should be selected for another growth experiment. In my opinion, the result will be more representative. There are, however a few inconsistencies that require further explanations or changes prior to publication in PLOS ONE.

The selection criteria of the broodstock were the same in the experiment. H5 and S. chuatsi were used as male and female parents respectively, and they were all mature adults with small growth differences. In the manuscript, we have added the description as the reviewer suggested. (see line 79-82)

In fact, size sorting were carried out before the growth experiment. The methods are as follows: when hybrid mandarin fish grow to 3 cm, we start domesticating them with artificial feed. The successful food transition took approximately 15 days. After that, the untamed individuals will be eliminated. When successfully domesticated individuals reached about 5.5-6.0cm, 600 individuals of the same size were selected to continue the growth experiment.(see line 84-90) 

Results:

Figure 1A and C: The result of 1A is not represented survival rate. The initial length of fish in Fig. 1C is between 5.69-5.92 cm, which is inconsistent of that mentioned in materials and methods.

As the reviewer mentioned, we have modified the Figure 1A. In addition, the ambiguous description in the MS may be puzzled the reviewer, the values (5.69-5.92 cm) in Fig 1C refers to the initial body length during the growth experiment, while the value (3 cm) in materials and methods refers to the body length at the beginning of domestication. Now, we have modified the descriptions in the corresponding section. (see line 84-90)

Figure 6: The calculation method of fold change in RNA-seq and qRT-PCR is different. It is suggested that the FC results of two analyses should be separated. By the way, the FC results of CDC20 are different in these two analyses. In addition, “RNA-sep” should be changed to “RNA-seq”.

As suggested, we have modified the Fig 6 and corrected the typo errors. The values of qRT-PCR is expressed the ratio of gene expression levels in the fast- growth group to those in the slow- growth group.

Furthermore, there are some typos in the indicated lines as follows:

Line 107, “De novo” should be italicized.

Line 132, There is an extra space between control and mainly.

Line 141, “Date” should be changed to “Data”.

As suggested, we have corrected the typos errors.

Reviewer #2: The manuscript (PONE-D-20-17403) entitled "Transcriptome analysis of the growth performance of hybrid mandarin fish after food conversion" which submitted to PLOS ONE by Guan WZ et al. has been reviewed.

The author used artificial feed to raise hybrid mandarin fish for 3 months, and measured the food conversion rate, survival rate, total length, and body weight of the hybrid fish, absolute length and weight gain rate. Then use RNA-Seq to identify potential genes involved in the pathways activated growth performance. For example: fatty acid biosynthesis (FASN and ACACB), collection of duct acid secretion (ATP6E and KCC4), cell cycle (CDC20 and CCNB) and insulin-like growth factor (insulin) -like growth factor, IGF) system (IGFBP1) and other pathways may be related to the growth of mandarin fish. Finally, it was also found in fast-growth fish, more potential single nucleotide polymorphisms (SNPs) were detected than in slow-growing fish. However, there are still some problems with the experimental design and content of this manuscript, which need to be supplemented and further explained and clarified by the author.

Q1-

What is the breeding pattern and origin of “the fifth generation hybrid fish” mentioned in this manuscript material and method? Regarding the source and definition of breeding biological materials, it is necessary to ask the author to add more details in the manuscript.

As suggested, we have added the breeding pattern and origin of “the fifth generation hybrid fish”in the MS. (see line75-79).

Q2-

What is the control group for the measurement of food conversion rate, survival rate, total length, body weight, absolute length, and weight gain rate of hybrid fish? In theory, there should be non-hybrid pure breeds or hybrids of different generations for comparison. To use this to emphasize the advantages of hybrids, the author should add Siniperca chuatsi and Siniperca scherzeri two uncrossed pure breeds. It is the control group (triple group) and compares various growth and survival values.

In our previous study, we have conducted the comparison experiments of hybrids F1 (S. scherzeri ♂× S. chuatsi ♀) and H6 (hybrid F5 ♂× S. chuatsi♀). The results showed that there was no significant difference in food conversion rate and survival rate (P > 0.05), however, the growth rate of H6 is 46.78% faster than that of F1 (P < 0.01) (Liu et al., 2019). In this study, we focus more on the growth differences in the hybrid fish after domestication, and the reason for this difference lies at the gene expression level.

Liu F, Guan W, Z., Wang C, L. Comparative analysis of domestication and growth of two hybrid mandarin fish. Fisheries Science and Technology Information. 2019; 46(6): 324-7.

Q3-

P156, Group1-3 in Figure 1(A) refers to the number of deaths at 0, 30, and 90 days in the triple-repeat population of the hybrid population. Why is there no 60-day data? It is impossible to understand why the author presented the triple-duplicate data of a hybrid species alone. What is the significance?

In fact, we counted the number of survivors every month, but the data of 60 days was missing in the data collation of the previous article . Thanks for your reminding, and now it has been added in Figure 1(A).

Q4-

Figure 1(C) and Figure 1(D) are the growth data of how many months old? It is suggested that the author should make up and give details.

Figure 1(C) and Figure 1(D) are the growth data of 0 day (Initial measurement) and 90 days (Final Measurement). AS suggested, we have added the details in MS (see line 166).

Q5-

In order to make it easier for readers to understand, basically the author should provide the growth data or growth curve of the F and S groups at each month of age?

As the reviewer mentioned, it would be nice to have the growth data at each month of age. But, the mandarin fish are timid and tend to hide. Each measurement can affect their eating situations in the next few days or even lead to death. Therefore, we only measured the pre-trial and post-trial growth data.

Q6-

Although the author mentioned in the manuscript that more potential single nucleotide polymorphisms (SNPs) are detected in fast-growth fish than in slow-growing fish. However, the author did not actually provide such comparative figures or tables in the manuscript.

As suggested, we have provided the corresponding table (See S4 table).

Q7-

There are more potential single nucleotide polymorphisms (SNPs) found in fast-growing fish. These nucleotide variation site markers are mainly located on those important growth genes. The authors also found in this result. No relevant research diagrams or tables are provided in this manuscript.

As suggested, we have provided the corresponding table (See S1 File).

---

## [Decision Letter · Decision Letter 1]

24 Sep 2020

Transcriptome analysis of the growth performance of hybrid mandarin fish after food conversion

PONE-D-20-17403R1

Dear Dr. Guan,

We’re pleased to inform you that your manuscript has been judged scientifically suitable for publication and will be formally accepted for publication once it meets all outstanding technical requirements.

Kind regards,

Tzong-Yueh Chen, Ph.D.

Academic Editor

PLOS ONE

Additional Editor Comments (optional):

Reviewers' comments:

Reviewer's Responses to Questions

**Comments to the Author**

1. If the authors have adequately addressed your comments raised in a previous round of review and you feel that this manuscript is now acceptable for publication, you may indicate that here to bypass the “Comments to the Author” section, enter your conflict of interest statement in the “Confidential to Editor” section, and submit your "Accept" recommendation.

Reviewer #1: All comments have been addressed

Reviewer #2: All comments have been addressed

2. Is the manuscript technically sound, and do the data support the conclusions?

Reviewer #1: Yes

Reviewer #2: Yes

3. Has the statistical analysis been performed appropriately and rigorously? 

Reviewer #1: Yes

Reviewer #2: Yes

4. Have the authors made all data underlying the findings in their manuscript fully available?

Reviewer #1: Yes

Reviewer #2: Yes

5. Is the manuscript presented in an intelligible fashion and written in standard English?

Reviewer #1: Yes

Reviewer #2: Yes

6. Review Comments to the Author

Reviewer #1: The authors have satisfactorily responded to my question and made the necessary changes to the manuscript. Thus, the revised version of the manuscript appears to be good for publishing in PLOS ONE.

Reviewer #2: The author has responded to related questions in the manuscript and actually provided supplementary information such as figures or tables.

7. PLOS authors have the option to publish the peer review history of their article (what does this mean?). If published, this will include your full peer review and any attached files.

Reviewer #1: No

Reviewer #2: No

---

## [Editor Report · Acceptance letter]

28 Sep 2020

PONE-D-20-17403R1 

Transcriptome analysis of the growth performance of hybrid mandarin fish after food conversion 

Dear Dr. Guan:

I'm pleased to inform you that your manuscript has been deemed suitable for publication in PLOS ONE. Congratulations! Your manuscript is now with our production department. 

Kind regards, 

on behalf of

Prof. Tzong-Yueh Chen 

Academic Editor

PLOS ONE